# Gut Microbiome in a Russian Cohort of Pre- and Post-Cholecystectomy Female Patients

**DOI:** 10.3390/jpm11040294

**Published:** 2021-04-12

**Authors:** Irina Grigor’eva, Tatiana Romanova, Natalia Naumova, Tatiana Alikina, Alexey Kuznetsov, Marsel Kabilov

**Affiliations:** 1Research Institute of Internal and Preventive Medicine—Branch of the Institute of Cytology and Genetics, Siberian Branch of Russian Academy of Sciences, Novosibirsk 630089, Russia; tarom_75@mail.ru; 2Institute of Chemical Biology and Fundamental Medicine, Siberian Branch of the Russian Academy of Sciences, Novosibirsk 630090, Russia; alikina@niboch.nsc.ru (T.A.); kabilov@niboch.nsc.ru (M.K.); 3Novosibirsk State Medical University, Novosibirsk 630091, Russia; 1xo2788353@mail.ru

**Keywords:** gallstone disease, 16S rDNA gene diversity, gut microbiota, blood biochemical characteristics

## Abstract

The last decade saw extensive studies of the human gut microbiome and its relationship to specific diseases, including gallstone disease (GSD). The information about the gut microbiome in GSD-afflicted Russian patients is scarce, despite the increasing GSD incidence worldwide. Although the gut microbiota was described in some GSD cohorts, little is known regarding the gut microbiome before and after cholecystectomy (CCE). By using Illumina MiSeq sequencing of 16S rRNA gene amplicons, we inventoried the fecal bacteriobiome composition and structure in GSD-afflicted females, seeking to reveal associations with age, BMI and some blood biochemistry. Overall, 11 bacterial phyla were identified, containing 916 operational taxonomic units (OTUs). The fecal bacteriobiome was dominated by *Firmicutes* (66% relative abundance), followed by *Bacteroidetes* (19%), *Actinobacteria* (8%) and *Proteobacteria* (4%) phyla. Most (97%) of the OTUs were minor or rare species with ≤1% relative abundance. *Prevotella* and *Enterocossus* were linked to blood bilirubin. Some taxa had differential pre- and post-CCE abundance, despite the very short time (1–3 days) elapsed after CCE. The detailed description of the bacteriobiome in pre-CCE female patients suggests bacterial foci for further research to elucidate the gut microbiota and GSD relationship and has potentially important biological and medical implications regarding gut bacteria involvement in the increased GSD incidence rate in females.

## 1. Introduction

Gallstone disease (GSD) has been, for many years, a significant public health problem worldwide, and its prevalence rate is expected to increase due to the ongoing changes in lifestyle and dietary habits. Gallstones are highly prevalent in Russia, with 100,000–200,000 cholecystectomies performed annually [1,2].

By now, there is little doubt about the multifaceted relationship between GSD and the microbiota [3], as some intestinal bacteria can promote gallstone formation [4,5,6], particularly by modifying the bile acid profile [7]. However, laparoscopic cholecystectomy (CCE), albeit currently a radical gold standard treatment, is not a neutral event and may increase the risk of some serious disorders and diseases, including metabolic syndrome, cardiovascular disease and cancers [4,8,9,10]. Post-cholecystectomy constant inflow of bile into the intestine and its metabolites can directly affect the intestinal microbiota [11], causing shifts in the gut–brain and gut–muscle axes and thus indirectly affecting the etiology and course of many related diseases and disorders. Although it is not yet possible to predict how particular perturbations will modify the microbiota, it is possible that different microbiome configurations might allow stratified treatment and diet recommendations in the future, becoming a novel and powerful candidate for personalized treatment of human diseases [4]. Among more than 500 oral drugs tested, 13% were discovered to be metabolized by the microbiome [12]. Another study identified 30 human gut microbiome-encoded enzymes responsible for the biotransformation of 20 drugs to 59 candidate metabolites [13]. Such findings strongly suggest the importance of including microbiomes into the framework of precision medicine. Although the pathogenesis of cholesterol gallstones is still not fully understood, gut microbiota dysbiosis plays an important role in their formation [14]. There is a paucity of studies describing fecal/gut bacteriobiome profiles in GSD-afflicted patients, both before and after surgery. The aim of our study was to inventory the fecal microbiota composition, as assessed by 16S rRNA gene sequencing, in a cohort of female patients with GSD and compare bacterial diversity before and after CCE.

## 2. Materials and Methods

### 2.1. Participants

Twenty-eight female patients with gallstone disease diagnosed by abdominal ultrasonography were recruited for the study (Table 1). The older patients had a higher BMI (Pearson’s correlation coefficient 0.66, *p* < 0.001). All patients underwent clinical examination to assess their gastrointestinal and gallbladder status and severity of their clinical condition; 21 patients had chronic disease, and the rest had acute disease. The patients had no history of treatment with antibiotics and proton pump inhibitors at least for 1 month prior to feces sampling, as well as no probiotics and/or prebiotics as special supplementation. Half of the patients had arterial hypertension, associated with increased BMI. The patients fasted for at least 12 h before the surgery. After the surgery, the patients received the antibiotic ceftriaxone. No specific diet was prescribed after the surgery.

All patients were duly informed, gave their consent to the study and signed the informed consent form. The study observed all the relevant institutional and governmental regulations. The protocol of the study was approved by the Ethic Committee of the Research Institute of Internal and Preventive Medicine-Branch of the Institute of Cytology and Genetics, SB RAS. All clinical aspects of the study were supervised by a gastroenterologist.

### 2.2. Fecal and Blood Sample Collection

Fecal samples were collected 1 day prior to the CCE and 1–3 days after the surgery, i.e., as soon as patients had stool, into 10 mL sterile fecal specimen containers and stored at −80 °C until use for DNA extraction. Blood samples were taken twice on the same day as stool samples.

### 2.3. Blood Analyses

Collected blood samples were used to determine aspartate aminotransferase (AST, EC 2.6.1.1) and alanine aminotransferase (ALT, EC 2.6.1.2) by the kinetic method, as recommended by the International Federation of Clinical Chemistry and Laboratory Medicine (IFCC2), using a biochemical analyzer, “Konelab Prime 30i” (Thermo Fisher Scientific, Vantaa, Finland).

### 2.4. Extraction of Total Nucleic Acid from Feces

Total DNA was extracted from 50 to 100 mg of thawed patient fecal samples using the MetaHIT protocol [15]. The bead beating was performed using TissueLyser II (Qiagen, Hilden, Germany), for 10 min at 30 Hz. No further purification of the DNA was needed. The quality of the DNA was assessed using agarose gel electrophoresis. No further purification of the DNA was needed.

### 2.5. 16S rRNA Gene Amplification and Sequencing

The 16S rRNA genes were amplified with the primer pair V3/V4, combined with Illumina adapter sequences [16]. PCR amplification was performed as described earlier [17]. A total of 200 ng of PCR product from each sample was pooled together and purified through MinElute Gel Extraction Kit (Qiagen, Hilden, Germany). The obtained amplicon libraries were sequenced with 2 × 300 bp paired-end reagents on MiSeq (Illumina, San Diego, USA) in the SB RAS Genomics Core Facility (ICBFM SB RAS, Novosibirsk, Russia). The read data reported in this study were submitted to GenBank under the study accession number PRJNA687360.

### 2.6. Bioinformatic and Statistical Analyses

Raw sequences were analyzed with the UPARSE pipeline [18] using Usearch v.11.0.667. The UPARSE pipeline included merging of paired reads; read quality filtering (-fastq_maxee_rate 0.005); length trimming (remove less than 350 nt); merging of identical reads (dereplication); discarding singleton reads; removing chimeras and OTU clustering using the UPARSE-OTU algorithm. The OTU sequences were assigned a taxonomy using SINTAX [19] on the RDP database. As a reference for bacteria, we used the 16S RDP training set v.16 [20]. Statistical analyses (descriptive statistics, Wilcoxon’s test for dependent variables, principal component analysis, multiple regression and general linear model, GLM, analysis with repeated measures) were performed by using Statistica v.13.3. Principle coordinate analysis (PCoA) was performed by PAST software v.3.17 [21]. The individual rarefaction showed that the sampling effort reached saturation for all samples (Appendix A); therefore, α-biodiversity indices were calculated for complete datasets using PAST software v.3.17 [21]. Statistical significance was defined as *p* < 0.05.

## 3. Results

### 3.1. Overall Bacteriobiome Diversity

After quality filtering and chimera and non-bacterial sequence removal, a total of 916 OTUs were identified at the 97% sequence identity level. All these OTUs could not be ascribed to a species level. Overall, they clustered into 11 phyla, 24 classes (with 18 explicitly classified at the class level), 59 orders (with 50 explicitly classified at the order level), 72 families (with 54 explicitly classified at the family level) and 172 genera (with 132 explicitly classified at the genus level). Sixty-one OTUs, i.e., ca. 7% of the species richness and ca. 0.1% of the relative abundance, could not be ascribed below the domain level. *Firmicutes* with 635 OTUs was, by far, the most species-rich phylum, accounting for 69% of the total number of OTUs. *Bacteroidetes* ranked second richest with 96 OTUs (10.5%), followed by *Actinobacteria* (64 OTUs, 7%) and *Proteobacteria* with 45 (5%). Such phyla as *Synergistetes*, *Tenericutes* and *Verrucomicrobia* were represented by 3–5 OTUs each, whereas the rest of the identified phyla, i.e., *Spirochaetes*, *Fusobacteria, Lentisphaerae* and cand. *Saccharibacteria*, were represented by one OTU each. The *Firmicutes* phylum was also the ultimate dominant phylum, accounting, on average, for ca. 66% of the total number of sequence reads. The *Firmicutes/Bacteroidetes* ratio varied widely: from 0.4 to 6837 (median 5.0) before CCE and from 0.4 to 2918 (median 3.8) after CCE (Appendix A), showing no CCE-related difference (*p* = 0.39, Wilcoxon’s test) and no correlation with blood biochemistry (Spearman’s, *p* > 0.05).

The dominant bacterial OTUs, i.e., OTUs contributing ≥ 1% (mean abundance) to the total number of sequence reads obtained for a sample, amounted to 27, with 18 OTUs representing *Firmicutes*, and *Bacteroidetes* and *Actinobacteria* contributing four and three OTUs, respectively, whereas *Veruccomicrobia* and *Synergestetes* each contributed one OTU to the dominants’ pool. Thus, the overwhelming majority (ca. 97%) of the OTUs in the study were minor or rare species.


### 3.2. Fecal Bacteriobiome Composition in GSD Patients before the CCE Surgery


The fecal microbiota of GSD patients was dominated by the *Firmicutes* phylum (66% relative abundance), followed by *Bacteroidetes* (19%), *Actinobacteria* (8%) and *Proteobacteria* (4%) phyla (Figure 1a). At the class level, the ultimate dominance of *Firmicutes* translated into the dominance of its classes *Clostridia* (47%), *Bacilli* (11%), *Negativicutes* (5%) and *Erysipelotrichia* (1.4%). *Bacteroidetes* was represented by the *Bacteroidia* class (19%), whereas *Actinobacteria* was represented solely by the *Actinobacteria* class. *Proteobacteria* was present mostly as *Gammaproteobacteria*, with *Alpha*-, *Beta*-, *Delta*- and *Epsilonproteobacteria* summarily contributing less than 1%. As for orders, *Clostridiales, Lactobacillales, Selenomonadales* and *Erysipelotrichales (Firmicutes), Bacteroidales (Bacteroidetes), Bifidobacteriales* and *Coriobacteriales (Actinobacteria)* and *Enterobacteriales (Gammaproteobacteria)* were found to prevail. Just three *Firmicutes* families, namely, *Ruminococcaceae*, *Lachnospiraceae* and *Enterococcaceae*, together accounted for half of the bacteriobiome abundance (Figure 1b). Most of the dominant OTUs (Figure 1c) belonged to the genera of the abovementioned families, i.e., *Enerococcus*, *Gemmiger*, *Faecalibacterium*, *Ruminococcus*, *Blautia*, *Roseburia* and *Streptococcus*. Other dominant OTUs represented *Bacteroidetes*/*Bacteroidia*/*Bacteroidales* (*Prevotella* sp. and *Bacteroides* sp.) and *Bifidobacteriaceae/Bifidobacteriales*/*Actinobacteria* (*Bifidobacterium* sp.).


Overall, fecal bacterial assemblages of GSD-afflicted subjects were characterized by high inter-individual variability of relative abundance and many outliers or extreme values at all taxonomic levels (Figure 1). Since we could not reasonably explain the outliers by errors in sampling collection and handling, nor by patients’ characteristics and analytical procedures, we performed principal component (PC) analysis (based on covariance) of the data matrix with bacterial relative abundances as variables for analysis and patients as subjects in order to (a) obtain a better insight into the variance structure throughout the cohort, (b) find an association of the major PCs with patients’ demographics and blood characteristics and then (c) implicate some bacterial taxa that contributed the most to the major PCs, as major players in such associations.

PC and multiple regression analyses showed that the core phyla, accounting for most of the data variance, showed a tendency for some association (PC2) with age, whereas some minor dominants (PC3, PC4) showed a correlation with blood glucose and bilirubin (Table 2).

As for the classes, the balance between the two core ones, i.e., *Clostridia* and *Bacteroidia* (PC2), was correlated with blood glucose, whereas the balance between two minor dominants, i.e., *Actinobacteria* and *Gammaproteobacteria* (PC4), was correlated with age (Appendix A).

The core orders, i.e., *Clostridiales* and *Lactobacillales*, both belonging to the different classes of the *Firmicutes* phylum, accounted for half of the data variance at this taxonomical level, showing some age correlation tendency (Appendix A).

At the family level, the balance between *Ruminococcaceae* and *Enterococcaceae* (PC1), both representing the ultimately dominating *Firmicutes* phylum, correlated strongly with the age and BMI of the studied cohort, whereas a tiny portion of the data variance (PC10), structured by the balance between *Veillonellaceae* and *Erysipelotrichaceae,* both also belonging to *Firmicutes*, was found to be associated with age, glucose and transaminase activity (Appendix A).

At the genus level, the relative abundance was structured mainly by the balance between *Prevotella* (*Bacteroidetes*) and *Enterococcus* (*Firmicutes*), PC1 showing a correlation with blood bilirubin (Table 3). The balance between *Faecalibacterium* and *Bifidobacterium* showed some association with glucose, whereas the relationship between *Blautia* and some unclassified genus of the *Ruminococcaceae* family (PC6) had a statistically significant correlation with blood bilirubin and transaminase activity (Table 3).

As for the species level, the major part of the relative abundance variance was accounted for by the relationship between *Enterococcus sp.* (*Firmicutes*) and *Prevotella copri* (*Bacteroidetes*), correlating with age, BMI and, possibly, blood glucose (Table 4), whereas small portions of the data variance, attributed to the balance between *Blautia luti* (*Firmicutes*) and *Akkermansia muciniphila* (PC6) and between two *Bifidobacterium* OTUs (PC8), could be partially ascribed to blood bilirubin and transaminase activity (Table 4).

### 3.3. Changes in Fecal Bacteriobiome Composition in GSD Patients after the CCE Surgery

At the phylum level, CCE did not show any effect, whereas an effect was revealed for the *Clostridia* class and the *Clostridiales* and *Coriobacteriales* orders (Table 5). Further down the taxonomical hierarchy, the effect was displayed by the differential surgery-related abundance of the *Clostridiaceae_1*, *Lachnospiraceae* and *Peptoniphilaceae* families (all belonging to *Clostridiales*) and *Coriobacteriaceae* of the namesake order of the *Actinobacteria* phylum (Table 6). *Lachnospiraceae*, being the predominating family with 129 OTUs in the studied cohort, accounted for 20% of the total number of *Firmicutes* OTUs and ranked the top family in abundance (with ca. 20%); its decreased post-CCE abundance was manifested by *Blautia, Roseburia* and some unclassified representatives of the family at the genus level (Table 6).

As for other genera, *Clostridium sensu stricto* increased, whereas *Peptoniphilus* decreased their presence in the fecal bacteriobiome of the studied cohort. Although *Coriobacteriaceae* increased their post-CCE abundance and were among the predominating families, at the genus level, they were represented by eight genera, only one of which (*Gordonibacter*) had a differential CCE-related abundance (*p* ≤ 0.05), albeit at the very low level, and *Collinsella* with its post-CCE increased abundance at the *p* ≤ 0.10 level (Table 6). At the species level, 20 OTUs manifested surgery-related differences in their relative abundance at the *p* ≤ 0.10 level of statistical significance, with 17 OTUs attributed to *Firmicutes*, two OTUs to *Actinobacteria* and one OTU to the *Bacteroidetes* phylum. *Enterococcus* sp. of *Firmicutes* was the leading dominant, increasing its abundance almost two-fold after the surgery (Table 6). *Bacteroides* sp. and *Collinsella* sp. were minor dominants with relative abundance around 1%. The rest of the OTUs with differential CCE-related abundance were minor or rare species.

The results of GLM analysis with repeated measures (before and after CCE) and age and BMI as continuous factors (covariates) show that residuals complied with a normal distribution only for the *Firmicutes* taxa; nevertheless, this statistical approach revealed no CCE-associated effect on the phylum and its lower taxa abundance.

### 3.4. Fecal Bacteriobiome α-Diversity before and after the CCE Surgery

No differences in α-diversity indices at the *p* ≤ 0.05 level were found in the studied cohort before and after CCE (Table 7). However, at the *p* ≤ 0.10 level, evenness slightly decreased, whereas the maximal relative abundance (as shown by the Berger–Parker index) slightly increased. The location of samples in the plane of the first two principal coordinates (based on Bray–Curtis distance) did not reveal any distinct CCE-related pattern (Figure 2).

### 3.5. Blood Biochemical Characteristics before and after the CCE Surgery

After CCE, both aspartate and alanine transaminase activity mean values increased 1.7 and 1.6 times, with bilirubin levels being unaffected (Table 8). Age and BMI as continuous predictors in GLM analysis with repeated measures decreased the *p*-value for glucose content comparison (*p* = 0.06) while increasing it for direct bilirubin content (*p* = 0.80) (residuals for only these two dependent variables showed a normal distribution).

## 4. Discussion

### 4.1. Fecal Bacteriobiome Composition in GSD Patients before the CCE Surgery

In the studied cohort of GSD-afflicted female patients, the *Firmicutes* phylum was the ultimate dominant in the fecal bacteriobiome, with *Bacteroidetes* and *Actinobacteria* being second and third in the ranking of abundance. In this aspect, our cohort differed from a Chinese one, where *Proteobacteria*, instead of *Actinobacteria*, were found to be third in abundance, and *Bacteroidetes* were almost twice as abundant as in our cohort [22]. Moreover, in our cohort, the *Firmicutes* relative abundance was markedly higher, as compared with the Chinese cohorts [5,21]. The apparent discrepancy is most likely due to the fact that one third of the Chinese cohort were males, whereas ours was a purely female one; and to the difference in diet [23]. As for the biodiversity indices, however, the ones calculated in our study agree well with the indices describing the fecal bacteriobiome of a Chinese cohort of GSD patients [21].

High inter-individual variation in bacterial sequence reads is quite common in fecal bacteriobiome studies in cohorts of healthy human subjects and of those afflicted by various diseases, including GSD [21]. Even in cases when the authors [21] claimed that their results “showed that the individual differences within the group were small”, the huge standard deviations for the OTUs’ relative abundance in their study proved the opposite. Therefore, by structuring down the data variance in our study by principal component analysis, we identified some age- and BMI-related taxa within the studied cohort, as well as some taxa that showed a correlation with blood glucose, bilirubin and transaminase activity.

Some genera of *Lachnospiraceae* are known to be important for bile acid metabolism, having 7α-dehydroxylation activity: in the pre-CCE bacteriobiome of the studied cohort, the family accounted for one fifth of the total number of sequence reads, along with *Ruminocccaceae* ultimately prevailing at the family level and together accounting for almost half of the abundance; in a Chinese cohort, however, the family with less than 0.3% was not even close to any dominating position [21]. The latter study also found the relative abundance of *Clostridium* to be 0.01%, whereas in our study, the presence of *Clostridium sensu stricto* was an order of magnitude more pronounced (Table 6). The differences are plentiful and most likely attributable to racial, dietary, sex and other characteristics of the cohorts.

Principal component analysis based on covariance allows dealing with the original variance of the data, without standardizing them, easily structuring the variance by extracted principal components, featuring the contribution of the original variables to the new ones (PCs) and reducing the original plethora of variables to fewer ones (PCs), but accounting for most of the original variance in the data. Subjection of the extracted PCs as dependent variables in multiple regression analysis allowed finding links with patients’ demographics and blood biochemical properties and bacterially interpreting them on the basis of a taxon contribution in the respective PC. We believe this approach to be informative for such kind of descriptive study.

The balance between two *Firmicutes* classes, namely, *Clostridia* and *Bacilli*, accounted for almost half of the abundance variance at this taxonomic level, being positively correlated with age at the *p* ≤ 0.10 level; the situation was translated in a similar manner at the order level, i.e., *Clostridiales* and *Lactobacillales*. Then, at the family level, the balance between *Ruminococcaceae* (*Clostridiales*) and *Enterococcaceae* (*Lactobacillales*), accounting for one third of the total data variance (PC1), displayed a statistically significant positive correlation with age. This finding complies with the knowledge that the gut microbiota diversity changes with age [24,25], with *Firmicutes* taxa increasing. Thus, the increased abundance of the core gut bacterial taxa with age in GSD-compromised subjects seems quite a natural occurrence, not overshadowed by changed bile acid metabolism and other factors. Interestingly, at the genus level, the structure of the data variance shifted, with the *Prevotella* (*Bacteroidetes*) and *Enterococcus* (*Firmicutes*) relationship accounting for half of the total data variance at this taxonomic level; the finding underscores the importance of these two core taxa relationships in structuring the gut bacteriome in general and GSD-compromised female patients in particular. As increased plasma levels of bilirubin (secondary to the breakdown of free hemoglobin) were shown to be associated with an increased risk of gallstone disease [26], the statistically significant positive correlation of the *Prevotella*–*Enterococcus* balance with blood bilirubin, found in our study, necessitates further investigation of the role of the genera in gallstone formation, both pigmented and cholesterol ones [27].

The finding that the balance between *Faecalibacterium, Bifidobacterium* and *Gemminer* determined 10% of the total data variance and was correlated with blood glucose (*p* ≤ 0.05) and BMI (*p* ≤ 0.10) may be indicative of the indirect association of the genera with glucose metabolism and insulin sensitivity in overweight patients [28], but we cannot currently suggest the putative cause–effect mechanism. The joint variation in *Blautia*, *Ruminococcus* and some unclassified *Ruminococcaceae* correlated with blood ALT, AST and bilirubin, and the mechanism of the involvement of these genera has to be comprehensively examined.

Further down the hierarchy at the species level, the *Prevotella*–*Enterococcus* relationship was manifested by the balance between *Prevotella copri* and *Enterococcus* sp. (with 40% of the total data variance), which was positively correlated with the cohort’s demographics, i.e., age and BMI (*p* ≤ 0.05), and glucose (*p* ≤ 0.10) and therefore may be related to glucose metabolism in overweight patients. Interestingly, *Akkermansia muciniphila* was found to contribute a small portion of the data variance associated with blood biochemistry (at *p* ≤ 0.05) and, hence, generally with the disease-compromised status of the subjects. The increased relative abundance of this mucin-degrading bacterium is often found to be associated with disease [29,30]; however, as a propionogenic bacterium, *A. muciniphila* is also believed to have several health benefits in humans [31].

However, as the blood characteristics are far from being specific for gallstone disease diagnostics, it is not possible to implicate the taxa in the changed bile acid metabolism and gallstone formation based on the results of multiple regression analysis, despite the comprehensive outline of the GSD bacteriobiome variation as related to the common blood biochemical properties. The bile acid profiles of the studied patients, if they had been determined, might have been more suggestive in this respect, and we acknowledge this as a drawback in the study.

### 4.2. Changes in Fecal Bacteriobiome Composition in GSD Patients after the CCE Surgery

In the studied cohort of GSD-afflicted female patients, mainly members of the *Firmicutes* phylum displayed CCE-related differential abundance.

As for the *Bacteroidetes* phylum, its relative abundance did not change after CCE; the result does not agree with the finding of Israeli researchers, for example, when the phylum abundance was shown to be increased in the post-CCE cohort of subjects [32]. However, the time factor, i.e., the duration of the time span elapsed between CCE and feces sampling, is a critical factor affecting bacterial composition [33] and may, at least partially, explain the discrepancy between results.

The decreased abundance of *Clostridia* (by 7%) and *Clostridiales* (by 3–5%), found after CCE in our study, was not reported before. As these taxa are the most species-rich and physiologically diverse components of the fecal bacteriobiome, the CCE-associated shifts in their relative abundance cannot be unequivocally regarded as beneficial or not for human health. *Lachnospiraceae*, the most predominant family in the human gut, displayed decreased abundance in the post-CCE bacteriobiome, which is, however, difficult to interpret as (a) most of the OTU-level clusters (69), ascribed to the family in our study (129), could not be taxonomically attributed below the family level, and (b) some genera and species of this family might support/contribute to healthy functions, whereas other genera and species were found to be increased in diseases [34]. For example, *Blautia* and *Roseburia* species, often associated with a healthy state, are some of the main short-chain fatty acid producers [35,36]; therefore, their post-CCE decreased abundance (*p* ≤ 0.10) may hardly be indicative of the better state of the gut bacteriobiome after surgery.

We could not find any information about the effect of CCE on *Actinobacteria*/*Coriobacteriales/Coriobacteriaceae* representatives, the latter known as pathobionts. As for *Collinsella*, the dominant genus in the *Coriobacteriaceae* family and the minor dominant in the studied cohort, its increased abundance after CCE (albeit at the *p* ≤ 0.10 level) suggests negative implications after such shift [37,38,39,40]. As for another representative of the family with differential pre- and post-CCE abundance, i.e., the *Gordonibacter* genus with just ca. 0.01% of the total number of sequence reads, it is difficult to suggest any ecophysiological significance at such abundance rate, although some genus representatives are known to participate in primary bile acid transformation [41] or be involved in dietary polyphenol transformations generating more bioactive metabolites [42].

Interestingly, although specific bacterial species such as *Helicobacter* and *Salmonella* were shown to be involved in the pathogenesis of cholesterol gallstones [43], we did not find *Helicobacter* at all, and found only one *Salmonella* OTU with 0.3 and 0.6% abundance in pre- and post-CCE subcohorts, respectively; the finding infers potentially different bacterial involvement in GSD etiology in cohorts of different sex and ethnicity.

The actual number of OTUs per sample observed in our study was practically the same as the number obtained by the same methodology for post-CCE patients in Korea [44]. However, as the latter study did not report whether the control group, i.e., non-CCE control patients, was also diagnosed with GSD, it is not possible to compare our results about the CCE effect on the fecal bacteriobiome with those results in their entirety, only for the post-CCE subcohort. For instance, in our study, CCE did not affect the gut bacteriobiome species richness, whereas compared with the independent control group, CCE decreased it [43]. Notably, the α-biodiversity index (Shannon) reported in the aforementioned study was much higher than the one reported here (4.9 vs. 2.8): in our view, the discrepancy may be attributed to both the sex composition of the Korean cohort and the dietary habits, etc. At the same time, for the Israeli cohort of GDS patients, the pre- and post-CCE values of the Shannon index did not differ [31], being close to, but slightly lower than, those in our study (2.1 vs. 2.8, respectively). We cannot help but emphasize that often the studies claiming to reveal the effect of CCE on the gut microbiota performed comparisons between the post-CCE patients with GSD and the healthy subjects without a GSD history [33,43,44]. We believe such approach does not seem to be adequate for aiming to examine the effect of CCE per se, as only a direct comparison between pre- and post-CCE conditions of one and the same cohort of GSD-affected subjects is pertinent for the goal of revealing microbiome shifts associated with the surgery and potential biomarkers of the latter.

It was shown that CCE did not markedly affect the bile acid profile in the GSD patients [31], leading the authors to conclude that the modified fecal bile acid composition results from inherently aberrant bile acid metabolism, leading, in turn, to gallstone formation. In general, our finding that the pre- and post-CCE fecal bacteriobiome profiles were not overall differentially distinct (as revealed by principal coordinate analysis) apparently complies with this conclusion. It should be emphasized that the repeated collection of feces samples was performed 1–3 days after the surgery, i.e., quite soon. Therefore, we are inclined to believe that it was a very short time to interpret the observed differential abundance of some bacterial taxa as solely resultant from the changed inflow of bile acids; the overall post-surgery condition most likely contributed significantly, if not primarily and predominantly, to the observed short-term CCE-related shifts in fecal bacterial assemblages. It should be noted that the overall post-surgery condition included ceftriaxone treatment of all patients. This beta-lactam antibiotic is able to kill a broad spectrum of bacteria [45], thus potentially shaping the gut bacteriobiome [46], especially when administered orally. Therefore, despite the very short time between the surgery and stool collection in this study, and hence the short time of antibiotic treatment, the revealed changes in the fecal bacteriobiome might have resulted, in part, due to the antibiotic per se. However, we should also emphasize that our study did not aim at discriminating between the effects of post-CCE altered bile profiles and antibiotic therapy; we aimed at profiling the gut bacteriobiome, referring to post-CCE as a single factor, as such embracing many factors, aspects, nuances, etc. We described the fecal bacteriobiome just at the starting point of patients embarking on the rest of life without a gallbladder. Whether the longer-term shifts in the gut microbiota after CCE will occur and to what degree and at what rate remain to be determined in future studies, which, hopefully, will also elucidate if gut microbes can act as the main character in the broad scenery of liver diseases [47].

## 5. Conclusions

Our study provides the first detailed inventory of the fecal bacteriobiome in a Russian cohort of female patients with gallstone disease. It will help to construct a global picture of the disease-related bacteriobiome and eventually focus on specific bacterial taxa involved in gallstone formation, thus facilitating the development of non-invasive therapeutic tools for preventing and treating gallstone disease. The shifts found in the fecal microbiota just a few days after CCE did not distinctly discriminate between the pre- and post-surgery bacterial diversity profiles. Therefore, the shifts can be mostly attributed to the surgery effect on the entire status of the patients, including the initial stages of the changing bile inflow and metabolism, as well as cellular and molecular modifications in the gut. The presented pre- and post-cholecystectomy microbiota profiles in one and the same cohort of patients may improve the insight into the relationship between the fecal, gut and bile microbiota, contributing to future larger-scale studies of altered human bile metabolism/profiles and associated disorders.

## Figures and Tables

**Figure 1 jpm-11-00294-f001:**
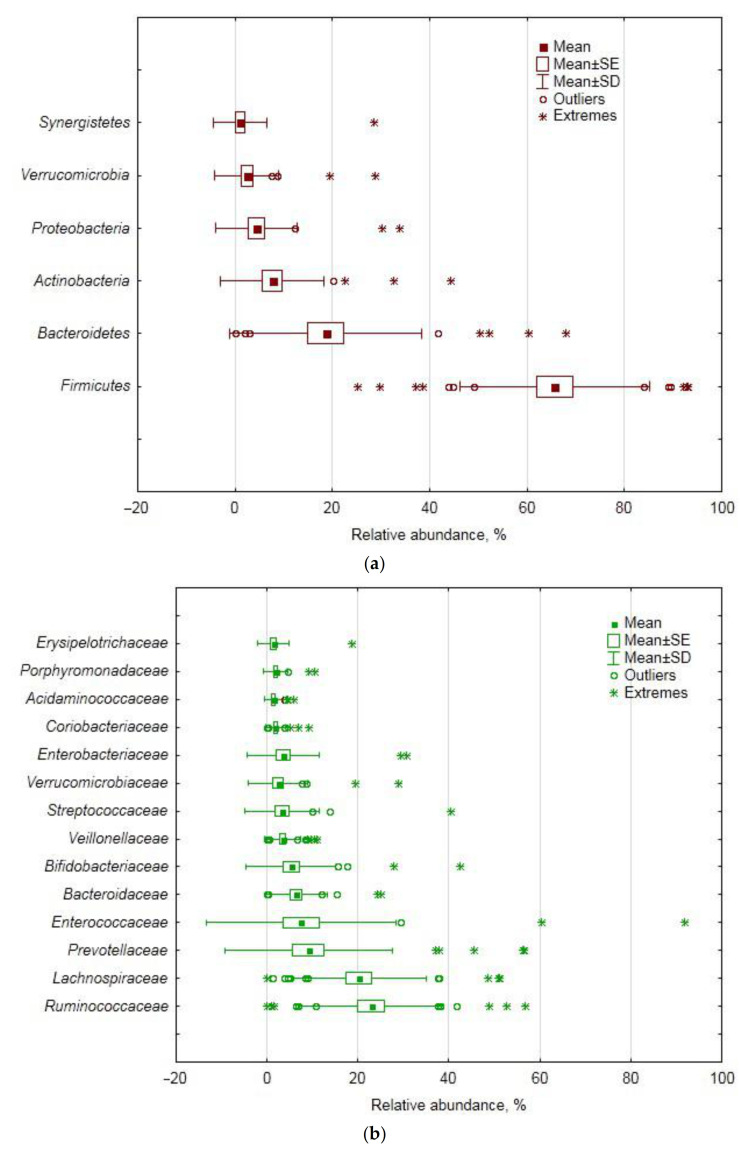
Relative abundance of the dominant bacterial taxa in females with gallstone disease (GSD) before cholecystectomy (CCE): (**a**) phyla, (**b**) families, (**c**) operational taxonomic units (OTUs).

**Figure 2 jpm-11-00294-f002:**
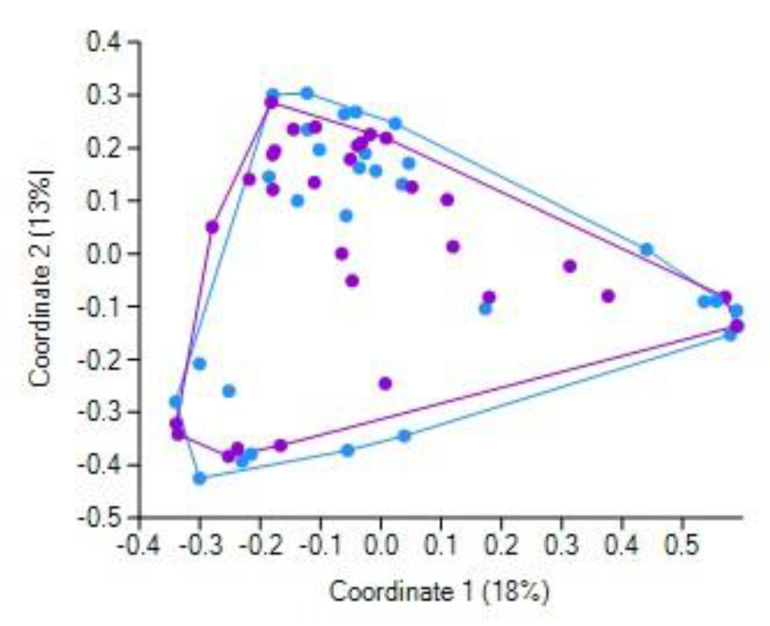
Location of patients’ samples taken before (violet markers) and after (blue markers) CCE in the plane of the first two principal coordinates (Bray–Curtis distance).

**Table 1 jpm-11-00294-t001:** Demographics of the study cohort (N = 28, females).

	Mean	Median	Min	Max
Age, years	51.6	55.0	18.0	73.0
BMI ^$^, kg/m^2^	25.7	24.8	17.6	34.5

^$^ BMI stands for body mass index.

**Table 2 jpm-11-00294-t002:** Statistical analyses’ results: contribution of bacterial phyla to the principal components extracted from the matrix with relative abundance in feces of females with GSD before CCE (percentage of the total data variance in brackets) and *p*-values for multiple regression with age, BMI and blood biochemistry.

**Phyla: PCA ^1^**				
Main	PC 1	PC 2	PC 3	PC 4
contributors	(65%)	(18%)	(11%)	(5%)
*Bacteroidetes*	**0.49**^2^ [−0.96] ^3^	**0.29** [−0.23]	0.04	0.01
*Firmicutes*	**0.48** [0.91]	**0.29** [−0.39]	0.00	0.06
*Proteobacteria*	0.01	0.14	**0.25** [0.03]	**0.14** [0.98]
*Actinobacteria*	0.01	**0.23** [0.97]	**0.59** [−0.18]	0.00
*Verrucomicrobia*	0.00	0.05	0.04	**0.78 [−0.13]**
**Phyla: Multiple regression**				
R/R^2^	0.54/0.29	0.53/0.28	0.61/0.37	0.57/0.33
Age	0.70	*0.09* ^4^	0.38	0.16
BMI	0.39	0.21	0.37	0.21
Glucose	0.13	1.00	0.07	0.70
ALT	0.84	0.68	0.11	0.12
AST	0.51	0.61	0.80	0.17
Bilirubin	0.13	0.20	0.70	*0.10*

^1^ PCA stands for principle component analysis (based on covariance). Only those principal components that (a) account for the bigger fraction of the total data variance and/or (b) displayed a statistically significant correlation with patients’ characteristics are shown. ^2^ The values in bold show the two topmost contributions. ^3^ Factor loadings for variables (taxon relative abundance) are given in square brackets. ^4^ The values in bold italics and underlined italics are at *p* ≤ 0.05 and *p* ≤ 0.10, respectively.

**Table 3 jpm-11-00294-t003:** Statistical analyses’ results: contribution of bacterial genera to the principal components extracted from the matrix with relative abundance in feces of females with gallstone disease before CCE (percentage of total variance in brackets) and *p*-values for multiple regression with blood biochemistry (coefficients of determination in brackets).

**Genera: PCA ^1^**				
Main	PC 1	PC 2	PC 5	PC 6
contributors	(35%)	(21%)	(5%)	(5%)
*Prevotella*	**0.10**^2^ [0.41] ^3^	**0.79** [−0.90]	0.00	0.00
*Enterococcus*	**0.84** [−0.97]	**0.07** [0.22]	0.01	0.00
*Faecalibacterium*	0.02	0.00	**0.17** [−0.37]	0.02
*Blautia*	0.00	0.03	0.01	**0.21** [0.50]
*Bifidobacterium*	0.01	0.04	**0.30** [−0.49]	0.00
*Gemmiger*	0.00	0.02	0.15 [0.45]	0.00
*Ruminococcus*	0.00	0.01	0.01	0.07 [−0.39]
un. *Ruminococcaceae*	0.00	0.01	0.01	**0.21** [−0.57]
**Genera: Multiple regression**				
R/R^2^	0.72/0.51	0.56/0.32	0.64/0.41	0.71/0.50
Age	0.16	0.92	0.13	0.12
BMI	0.29	0.72	*0.09 * ^3^	0.42
Glucose	0.71	0.25	***0.03*** ^4^	0.76
ALT	0.55	0.22	0.12	***0.03***
AST	0.73	0.48	*0.10*	***0.03***
Bilirubin	***0.04***	0.55	0.17	***0.04***

^1^ PCA stands for principle component analysis (based on covariance). Only those principal components that (a) account for the bigger fraction of the total data variance and/or (b) displayed a statistically significant correlation with patients’ characteristics are shown. ^2^ The values in bold show the two topmost contributions. ^3^ Factor loadings for variables (taxon relative abundance) are given in square brackets. ^4^ The values in bold italics and underlined italics are at *p* ≤ 0.05 and *p* ≤ 0.10, respectively.

**Table 4 jpm-11-00294-t004:** Statistical analyses’ results: contribution of bacterial OTUs into the principal components extracted from the matrix with relative abundance in feces of females with gallstone disease before CCE (percentage of total variance in brackets) and *p*-values for multiple regression with age, BMI and blood biochemistry (coefficients of determination in brackets).

**OTUs: PCA ^1^**								
Main	PC 1	PC 3	PC 4		PC 6		PC 8	
contributors	(40%)	(9%)	(7%)		(3%)		(2%)	
*Enterococcus sp.*	**0.86**^2^ [0.96] ^3^	0.00	0.00		0.00		0.00	
*Escherichia/Shigella* sp.	0.00	**0.25** [−0.65]	**0.25** [−0.57]		0.07		0.01	
*Gemmiger*	0.00	0.01	**0.28** [0.67]		0.00		0.00	
*Faecalibacterium prausnitzii*	0.01	**0.63** [0.86]	0.16 [−0.38]		0.06		0.02	
*Akkermansia muciniphila*	0.00	0.02	0.09 [0.41]		**0.22** [−0.45]		0.09	
*Blautia luti*	0.00	0.01	0.05		**0.26** [−0.56]		0.02	
*Bifidobacterium* sp.	0.00	0.03	0.02		0.07 [−0.30]		**0.35** [−0.49]	
*Bifidobacterium* sp.	0.00	0.00	0.00		0.19 [0.60]		**0.27** [−0.52]	
*Streptococcus* sp.	0.00	0.00	0.00		0.02		0.13 [0.48]	
*Prevotella copri*	**0.12** [−0.42]	0.00	0.00		0.01		0.00	
**OTUs: Multiple regression**								
R/R^2^	0.71/0.50	0.39/0.15		0.40/0.16		0.73/0.53		0.54/0.29
Age	***0.05*** ^4^	0.16		0.34		0.55		0.11
BMI	***0.03***	*0.10*		0.92		*0.09*		0.35
Glucose	*0.07 * ^3^	0.58		*0.07*		0.39		0.76
ALT	0.52	0.65		0.92		***0.00***		***0.03***
AST	0.68	0.74		0.98		***0.00***		***0.03***
Bilirubin	0.16	0.95		0.64		***0.02***		***0.05***

^1^ PCA stands for principle component analysis (based on covariance). Only those principal components that (a) account for the bigger fraction of the total data variance and/or (b) displayed a statistically significant correlation with patients’ characteristics are shown. ^2^ The values in bold show the two topmost contributions. ^3^ Factor loadings for variables (taxon relative abundance) are given in square brackets. ^4^ The values in bold italics and underlined italics are at *p* ≤ 0.05 and *p* ≤ 0.10, respectively.

**Table 5 jpm-11-00294-t005:** The relative abundance of some higher bacterial taxa in patients’ feces before and after CCE.

	Before CCE		After CCE		*p*-Value
	Median	Mean ± SD	Median	Mean ± SD	
Phyla (dominant)					
*Firmicutes*	66.7	65.7 ± 19.6	64.7	65.4 ± 21.4	0.34
*Bacteroidetes*	12.2	18.6 ± 19.8	11.0	19.2 ± 21.4	0.53
*Actinobacteria*	3.8	7.7 ± 10.6	4.0	7.2 ± 9.8	0.96
*Proteobacteria*	0.5	4.4 ± 8.4	0.8	2.8 ± 4.9	0.55
*Verrucomicrobia*	0.0	2.4 ± 6.6	0.0	3.8 ± 9.4	0.09
un. ^1^ *Bacteria*	0.1	0.14 ± 0.27	0.1	0.12 ± 0.23	0.20
Classes (dominant)					
*Clostridia* ^2^	49.3	46.8 ± 23.1	42.9	40.7 ± 23.1	0.01
*Bacteroidia*	12.2	18.6 ± 19.8	10.9	19.2 ± 21.4	0.47
*Bacilli*	1.2	11.1 ± 23.0	0.5	16.8 ± 30.0	0.51
*Actinobacteria*	3.8	7.7 ± 10.6	4.0	7.2 ± 9.8	0.97
*Negativicutes*	4.5	4.9 ± 3.9	3.0	5.2 ± 6.3	0.84
*Verrucomicrobiae*	0.0	2.4 ± 6.6	0.0	3.8 ± 9.4	0.09
*Gammaproteobacteria*	0.0	3.5 ± 8.2	0.0	2.1 ± 5.0	0.43
un. *Firmicutes*	0.3	1.5 ± 3.2	0.5	1.4 ± 1.9	0.70
*Erysipelotrichia*	0.5	1.4 ± 3.5	0.4	1.4 ± 2.6	0.29
Orders (dominant)					
*Clostridiales*	43.3	44.8 ± 23.5	40.5	39.0 ± 23.2	0.01
*Bacteroidales*	12.5	20.4 ± 20.4	14.6	21.3 ± 22.2	0.47
*Lactobacillales*	1.2	12.0 ± 22.5	0.9	16.9 ± 28.9	0.50
*Selenomonadales*	4.5	5.2 ± 4.5	3.2	5.7 ± 6.7	0.84
*Bifidobacteriales*	0.7	4.9 ± 9.6	0.0	4.4 ± 8.8	0.53
*Verrucomicrobiales*	0.0	2.3 ± 6.4	0.0	3.5 ± 9.1	0.09
*Enterobacteriales*	0.0	3.3 ± 7.7	0.0	1.9 ± 4.7	0.39
*Coriobacteriales*	0.8	1.9 ± 2.2	1.0	2.4 ± 2.7	0.03
*Erysipelotrichales*	0.5	1.4 ± 3.4	0.4	1.3 ± 2.5	0.25

^1^ un. stands for unclassified. ^2^ Gray-shadowed lines have *p*-values ≤ 0.05.

**Table 6 jpm-11-00294-t006:** The relative abundance of some bacterial families, genera and OTUs in patients’ feces before and after CCE.

Taxon	Before CCE		After CCE		*p*-Value
	Median	Mean ± SD	Median	Mean ± SD	
Families					
*Clostridiaceae_1* ^4^		0.2 ± 0.4		0.3 ± 0.5	0.010
*Lachnospiraceae*		20.3 ± 14.8		16.6 ± 14.0	0.032
*Coriobacteriaceae*		1.9 ± 2.3		2.4 ± 2.8	0.033
*Peptoniphilaceae*		0.005 ± 0.013		0.002 ± 0.007	0.036
*Peptostreptococcaceae*		0.4 ± 0.6		0.6 ± 1.0	0.054
*Ruminococcaceae*		22.9 ± 15.9		19.5 ± 13.8	0.056
*Rhodospirillaceae*		0.1 ± 0.4		0.01 ± 0.05	0.059
*Enterococcaceae*		7.4 ± 20.8		13.6 ± 26.4	0.080
Genera					
*Clostridium s.s.*		0.2 ± 0.4		0.3 ± 0.5	0.010
*Gordonibacter*		0.01 ± 0.04		0.01 ± 0.07	0.036
*Peptoniphilus*		0.005 ± 0.013		0.002 ± 0.007	0.036
un ^1^. *Rhodospirillaceae*		0.08 ± 0.36		0.01 ± 0.05	0.059
*Gemmiger*		4.3 ± 7.4		3.2 ± 4.8	0.065
*Collinsella*		1.0 ± 1.8		1.3 ± 2.1	0.066
*Blautia*		6.4 ± 8.8		4.9 ± 7.3	0.067
*Enterococcus*		7.4 ± 20.8		13.6 ± 26.4	0.080
*Faecalibacterium*		7.7 ± 9.8		6.1 ± 8.6	0.089
*Roseburia*		2.8 ± 4.9		1.6 ± 2.5	0.089
*Dialister*		1.5 ± 2.5		1.0 ± 2.1	0.093
*Peptostreptococcus*		0.018 ± 0.063		0.024 ± 0.077	0.093
*Lachnospiracea i.s.*		2.1 ± 2.0		1.5 ± 1.6	0.094
OTUs					
un.*Lachnospiraceae*		0.09 ± 0.17		0.04 ± 0.11	0.003
un.*Clostridium s.s.* ^2^		0.14 ± 0.43		0.3 ± 0.5	0.005
un.*Clostridium XlVa*		0.01 ± 0.03		0.03 ± 0.05	0.021
un.*Clostridiales*		0.02 ± 0.06		0.09 ± 0.19	0.023
*Clostridium leptum*		0.05 ± 0.10		0.03 ± 0.07	0.025
un.*Blautia*		0.02 ± 0.06		0.03 ± 0.08	0.029
*Ruminococcus faecis*		0.4 ± 1.4		0.6 ± 1.5	0.030
un.*Bacteroides*		2.1 ± 3.4		1.0 ± 2.0	0.033
*Dialister invisus*		1.0 ± 2.1		0.4 ± 1.0	0.035
un.*Ruminococcus*		0.15 ± 0.37		0.07 ± 0.24	0.036
un.*Ruminococcus*		0.2 ± 0.6		0.07 ± 0.29	0.042
un.*Lachnospiracea* *i.s.* ^3^		0.5 ± 0.8		0.2 ± 0.4	0.050
un.*Coriobacteriaceae*		0.04 ± 0.09		0.14 ± 0.40	0.059
un.*Ruminococcaceae*		0.09 ± 0.16		0.18 ± 0.30	0.068
un.*Ruminococcaceae*		0.0008 ± 0.0017		0.0002 ± 0.001	0.076
un.*Enterococcus*		7.4 ± 20.8		13.6 ± 26.4	0.080
un.*Lachnospiraceae*		0.01 ± 0.02		0.002 ± 0.007	0.080
un.*Clostridiales*		0.01 ± 0.03		0.03 ± 0.08	0.083
un.*Collinsella*		1.0 ± 1.8		1.3 ± 2.1	0.093
*Peptostreptococcus stomatis*		0.02 ± 0.06		0.02 ± 0.08	0.093

^1^ un. stands for unclassified; ^2^
*s.s.* stands for *sensu stricto*; ^3^
*i.s*. stands for *incertae sedis*. ^4^ Gray-shadowed lines have *p*-values ≤ 0.05.

**Table 7 jpm-11-00294-t007:** Alpha-diversity indices estimated for patients’ fecal bacterial assemblages before and after CCE.

Index	Before CCE		After CCE		*p*-Value
	Median	Mean ± SD	Median	Mean ± SD	
OTUs’ richness	90	93 ± 41	78	92 ± 46	0.89
Chao-1	100	104 ± 51	85	104 ± 55	0.75
Berger–Parker	0.24	0.29 ± 0.19	0.27	0.33 ± 0.21	0.07
Dominance (D)	0.12	0.16 ± 0.16	0.12	0.19 ± 0.17	0.11
Simpson (1-D)	0.88	0.84 ± 0.16	0.88	0.81 ± 0.17	0.11
Shannon	2.8	2.8 ± 0.8	2.7	2.7 ± 0.9	0.21
Evenness	0.23	0.22 ± 0.08	0.20	0.20 ± 0.09	0.09
Equitability	0.67	0.63 ± 0.13	0.62	0.60 ± 0.15	0.13

**Table 8 jpm-11-00294-t008:** Blood biochemical test results before and after CCE.

Property	Before CCE		After CCE		*p*-Value
	Median	Mean ± SD	Median	Mean ± SD	
Glucose, mmol/L	5.25	5.24 ± 0.88	5.45	5.51 ± 0.82	0.107
Alanine transaminase, U/L	20.2	36.0 ± 68.5	29.0	57.4 ± 75.7	0.001
Aspartate transaminase, U/L	19.9	34.4 ± 47.9	32.3	56.8 ± 68.4	0.001
Bilirubin total, mcmol/L	15.1	18.8 ± 21.1	16.3	18.4 ± 7.0	0.168
Bilirubin direct, mcmol/L	11.8	12.4 ± 6.0	13.3	12.7 ± 5.7	0.186
Bilirubin indirect, mcmol/L	3.5	6.4 ± 16.7	3.6	3.7 ± 1.7	0.190

## Data Availability

The read data reported in this study were submitted to GenBank under the study accession number PRJNA687360 (https://www.ncbi.nlm.nih.gov/bioproject/?term=PRJNA687360).

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
