# Peer review of "Gut Microbiome in a Russian Cohort of Pre- and Post-Cholecystectomy Female Patients"

_jpm, 2021, doi:10.3390/jpm11040294_

Round 1
Reviewer 1 Report
Introduction: In the introduction I deeply suggest citing this two recent works that explain in detail every basic (up-to-date) knowledge that you need to know about microbiota: 1 - doi: https://doi.org/10.1152/ajpgi.00161.2019 | 2 - doi: https://doi.org/10.3390/jcm9113705
M&M:
- Participants information should be reported in the results section as well as correlation findings;
- Please define statistical significance
- How was multiple linear regression studied? How were variables included in the model? How was the best model chosen? (please, check and cite the statistical part of this study published in another MDPI journal: DOI: 10.3390/diagnostics10090619
Author Response
Response to Reviewer 1 Comments
|
# |
Introduction: |
|
Point 1 |
In the introduction I deeply suggest citing this two recent works that explain in detail every basic (up-to-date) knowledge that you need to know about microbiota: 1 - doi: https://doi.org/10.1152/ajpgi.00161.2019 | 2 - doi: https://doi.org/10.3390/jcm9113705 |
|
Response 1 |
There is no denying that it can be tremendously helpful when reviewers suggest taking into consideration some publications pertaining to the submitted manuscript. So we carefully read both articles. We added the text “which hopefully will also elucidate if gut microbes can act as the main character in the broad scenery of liver diseases [45]” into the manuscript and the relevant reference (the first one) into the reference list. The changes are highlighted in turquoise in the revised version of our manuscript. As for the second recommended article, albeit very exciting and detailed, it is much broader in the topic (as well as animal subjects), so, in order not to overburden the text and the reference list, we decided to refer this article in our subsequent work (on fecal microbiome in patents with multiple sclerosis where speculations about ENS, CNS and microbiota associations could be very interesting and perfectly relevant). |
|
|
M&M: |
|
Point 2 |
Participants information should be reported in the results section as well as correlation findings; |
|
Response 2 |
We are afraid that, like many other researchers, we believe the information about patients belongs to the M&M section, therefore saw no need to change it. |
|
Point 3 |
Please define statistical significance. |
|
Response 3 |
We added the text: “Statistical significance was defined as p < 0.05”. The changes are highlighted in turquoise in the revised version of our manuscript. |
|
Point 4 |
How was multiple linear regression studied? How were variables included in the model? How was the best model chosen? (please, check and cite the statistical part of this study published in another MDPI journal: DOI: 10.3390/diagnostics10090619 |
|
Response 4 |
As indicated in the M&M, we used rather straightforwardly Statistica v. 13.3 software module for linear regression. All available variables were included. We checked the recommended paper; unfortunately, there is no way we can cite it as to do so we should have the 1) SPSS software, as specific analyses’ modules are not similar in each and every package in order to be able to have it at one’s fingertips; and 2) we simply had no time (within the given time frame to submit a revised version of our manuscript) to get familiar with the approach employed in the recommended paper. Because of these we cannot possibly currently cite the paper, but shall remain to be seized of the matter. |

Reviewer 2 Report
in line 34 cancel as well
lines 38-39-40 re-formulate the sentence.
The remanent part sounds good!
I think that paper could be very interesting for biology and microbiology but not too much for clinicians. To make more interesting to clinical readers i could suggest to the authors to add some articles like:
1-Longhitano Y, Zanza C, Thangathurai D, Taurone S, Kozel D, Racca F, Audo A, Ravera E, Migneco A, Piccioni A, Franceschi F. Gut Alterations in Septic Patients: A Biochemical Literature Review. Rev Recent Clin Trials. 2020;15(4):289-297. doi: 10.2174/1574887115666200811105251. PMID: 32781963.
2-Zanza C, Thangathurai J, Audo A, Muir HA, Candelli M, Pignataro G, Thangathurai D, Cicchinelli S, Racca F, Longhitano Y, Franceschi F. Oxidative stress in critical care and vitamins supplement therapy: "a beneficial care enhancing". Eur Rev Med Pharmacol Sci. 2019 Sep;23(17):7703-7712. doi: 10.26355/eurrev_201909_18894. PMID: 31539163.
You should explain the role of oxitadive stress in sepsis related to acute cholecystitis or the gut alteration after and before surgery related to CCE.
Please can you resend me when you had corrections?
Author Response
Response to Reviewer 2 Comments
|
Point 1 |
in line 34 cancel as well
|
|
Response 1 |
We removed “as well” as recommended; the change is highlighted in turquoise in the revised version of the manuscript. |
|
Point 2 |
lines 38-39-40 re-formulate the sentence. |
|
Response 2 |
We reformulated the sentence by splitting in two ones, and combining the references; the change is highlighted in turquoise in the revised version of the manuscript. |
|
Point 3 |
The remaining part sounds good! |
|
Response 3 |
Thank you very much for the praise! |
|
Point 4 |
I think that paper could be very interesting for biology and microbiology but not too much for clinicians. To make more interesting to clinical readers I could suggest to the authors to add some articles like:
1-Longhitano Y, Zanza C, Thangathurai D, Taurone S, Kozel D, Racca F, Audo A, Ravera E, Migneco A, Piccioni A, Franceschi F. Gut Alterations in Septic Patients: A Biochemical Literature Review. Rev Recent Clin Trials. 2020;15(4):289-297. doi: 10.2174/1574887115666200811105251 2-Zanza C, Thangathurai J, Audo A, Muir HA, Candelli M, Pignataro G, Thangathurai D, Cicchinelli S, Racca F, Longhitano Y, Franceschi F. Oxidative stress in critical care and vitamins supplement therapy: "a beneficial care enhancing". Eur Rev Med Pharmacol Sci. 2019 Sep;23(17):7703-7712. doi: 10.26355/eurrev_201909_18894. You should explain the role of oxidative stress in sepsis related to acute cholecystitis or the gut alteration after and before surgery related to CCE. |
|
Response 4 |
We ultimately agree that everything is tightly intertwined and interconnected in a human body, and we read with great interest the recommended articles; however exciting and useful they are, their topics do not seem to be immediately pertinent to the main subject and message of our study, therefore our imagination concerning where exactly we could refer to these articles failed us, unfortunately, as we could not possibly figure out how to embrace the entire plethora of medical problems within a discussion in one article. |

Reviewer 3 Report
Grigor’eva et al. investigated the gut microbiota difference before and after cholecystectomy in female Russian patients. Several study concerns should be mentioned.
1. Different α-diversity indices were calculated in this study, including OTUs’ richness, Chao 1, Berger-Parker, Dominance, Simpson, Shannon, Evenness, and Equitability. However, the detailed methodology and the difference between α-diversity indices were not mentioned in the method section.
2. The OTU sequences were assigned taxonomy using the SINTAX. However, the available databases were not mentioned, such as the RDP database (RDPDB), SILVA, Greengenes, or UNITE.
3. There are too many tables in the main text. Please remove some tables to supplementary files. Please focus on key messages to illustrate the microbiota change before and after cholecystectomy.
4. Rarefaction curves based on gene count were not evaluated first to prevent methodological artifacts originating from variations in sequencing depth.
5. This study was small sample sizes, so the results were difficult to interpret.
6. No p-value was shown in β-diversity (Brey-Curtis distance, Figure 1).
7. References 43 and 45 are duplicated.
Author Response
Response to Reviewer 3 Comments
|
Point 1 |
Different α-diversity indices were calculated in this study, including OTUs’ richness, Chao 1, Berger-Parker, Dominance, Simpson, Shannon, Evenness, and Equitability. However, the detailed methodology and the difference between α-diversity indices were not mentioned in the method section. |
|
Response 1 |
We are afraid we believe that describing detailed methodology in this manuscript would be far beyond the scope of the study and its methodology; the more so that the reference to it is provided in M&M section. Moreover, since there were no statistically significant differences, the detailed methodology description does not seem particularly relevant, does it? |
|
Point 2 |
The OTU sequences were assigned taxonomy using the SINTAX. However, the available databases were not mentioned, such as the RDP database (RDPDB), SILVA, Greengenes, or UNITE. |
|
Response 2 |
Although the RDP was mentioned in M&M section, we put it more explicitly by adding the following phrase “ on the RDP data base” after mentioning SINTAX in the M&M; the addition is highlighted in turquoise in the revised version of the manuscript. |
|
Point 3 |
There are too many tables in the main text. Please remove some tables to supplementary files. Please focus on key messages to illustrate the microbiota change before and after cholecystectomy. |
|
Response 3 |
We removed three Tables to Supplementary Material. |
|
Point 4 |
Rarefaction curves based on gene count were not evaluated first to prevent methodological artifacts originating from variations in sequencing depth. |
|
Response 4 |
We are sorry we did not mention the rarefaction analysis. So we added the appropriate information in the text (The individual rarefaction showed that the sampling effort reached saturation for all samples (Figure S1), therefore α-biodiversity indices were calculated for complete data sets using PAST software v.3.17 [21]; the text is highlighted in turquoise in the revised version), and the corresponding graph with the rarefaction curve was added in Supplementary Material. |
|
Point 5 |
This study was small sample sizes, so the results were difficult to interpret. |
|
Response 5 |
There are plenty of published fecal microbiome studies with the sample sizes much smaller than the one presented in our manuscript. Almost 30 patients with repeated measures comprise rather representative cohort for microbiome descriptional profiling. |
|
Point 6 |
No p-value was shown in β-diversity (Brey-Curtis distance, Figure 1). |
|
Response 6 |
The results of the Principal Coordinates Analyses presented on the graph just illustrate the structure of relationship without any p-values. |
|
Point 7 |
References 43 and 45 are duplicated. |
|
Response 7 |
Extra reference removed, and the reference number in the text corrected; the change in the revised version is highlighted in turquoise. |

Round 2
Reviewer 1 Report
The paper is much improved now.
Author Response
Thank you very much! We do really appreciate the time you spent on reviewing our manuscript!
Reviewer 2 Report
Thank You for your excellent revision.
i think that at least this one could be help for you......but it is not mandatory.
Longhitano Y, Zanza C, Thangathurai D, Taurone S, Kozel D, Racca F, Audo A, Ravera E, Migneco A, Piccioni A, Franceschi F. Gut Alterations in Septic Patients: A Biochemical Literature Review. Rev Recent Clin Trials. 2020;15(4):289-297. doi: 10.2174/1574887115666200811105251. PMID: 32781963.
Anyway now it is ready to publish for me
Author Response
Actually, we also think so, and most likely will do this in our further publications as we have planned a series of articles about fecal microbiome alterations in patients with various gastrointestinal diseases. The main reason we did not cite it right now is the pressure of deadlines for revisions, and we did not want to cite the suggested article haphazardly and hence, strictly speaking, compromising all the parties involved. Anyway, thank you very much for all your comments and suggestions! We do really appreciate the time you spent on reviewing our manuscript!

Reviewer 3 Report
All comments had been replied. I have no further suggestions.
Author Response
Thank you very much! We do rally appreciate the time you spent to review our manuscript!